# Inhibition of LNC EBLN3P Enhances Radiation-Induced Mitochondrial Damage in Lung Cancer Cells by Targeting the Keap1/Nrf2/HO-1 Axis

**DOI:** 10.3390/biology12091208

**Published:** 2023-09-04

**Authors:** Haoyi Tang, Shanghai Liu, Xiangyu Yan, Yusheng Jin, Xiangyang He, Hao Huang, Lu Liu, Wentao Hu, Anqing Wu

**Affiliations:** 1State Key Laboratory of Radiation Medicine and Protection, School of Radiation Medicine and Protection, Collaborative Innovation Center of Radiological Medicine of Jiangsu Higher Education Institutions, Soochow University, Suzhou 215123, China; 2Suzhou Medical College, Soochow University, Suzhou 215123, China

**Keywords:** lung cancer, long non-coding RNA, radiotherapy, radiosensitivity, mitochondrial damage

## Abstract

**Simple Summary:**

The intrinsic radioresistance of cancer cells is a major barrier to effective radiotherapy for NSCLC. The long non-coding RNA, endogenous bornavirus-like nucleoprotein 3 (LNC EBLN3P), was initially discovered to exhibit upregulation in NSCLC and promote the progression of NSCLC. The aim of this study was to elucidate the correlation between elevated expression levels of LNC EBLN3P and radioresistance in NSCLC cells, as well as to investigate the underlying molecular mechanism. We confirmed that the inhibition of LNC EBLN3P can enhance the production of reactive oxygen species (ROS) and trigger mitochondrial impairment in NSCLC cells, thereby mitigating the radioresistance of NSCLC cells. Therefore, the inhibition of LNC EBLN3P presents a promising approach for the development of novel radiosensitizers.

**Abstract:**

Lung cancer remains the leading cause of cancer-related deaths in both women and men, claiming millions of lives worldwide. Radiotherapy is an effective modality for treating early-stage lung cancer; however, it cannot completely eradicate certain tumor cells due to their radioresistance. Radioresistance is commonly observed in conventionally fractionated radiotherapy, which can lead to treatment failure, metastasis, cancer recurrence, and poor prognosis for cancer patients. Identifying the underlying molecular mechanisms of radioresistance in lung cancer can promote the development of effective radiosensitizers, thereby improving patients’ life expectancy and curability. In this study, we identified LNC EBLN3P as a regulator of lung cancer cell proliferation and radiosensitivity. The repression of LNC EBLN3P could increase ROS production and mitochondrial injury in NSCLC cells. In addition, knocking down LNC EBLN3P increased the binding of Nrf2 to Keap1, resulting in enhanced Nrf2 degradation, decreased translocation of Nrf2 to the nucleus, reduced expression of antioxidant protein HO-1, weakened cellular antioxidant capacity, and increased radiosensitivity of NSCLC cells. These findings suggest that targeting LNC EBLN3P could be a promising strategy for developing novel radiosensitizers in the context of conventional radiotherapy for NSCLC.

## 1. Introduction

Lung cancer is one of the most common fatal cancers in the world. According to the latest data, the death rate for lung cancer in 2023 is expected to account for 21% of all cancer deaths, which seriously affects the quality of life and health of people around the world [1]. Lung cancer is divided into two main categories: small-cell lung cancer (SCLC) and non-small-cell lung cancer (NSCLC). NSCLC has a relatively high incidence, accounting for approximately 85% of all cases of lung cancer [2]. Improving the survival rate of NSCLC patients has become the focus of lung cancer prevention and treatment. The main treatments for NSCLC include surgery, chemotherapy, and radiotherapy [3]. Unfortunately, most lung cancer patients are diagnosed in the middle and late stages, with a survival rate of less than one year. Therefore, radiotherapy is often preferred as a treatment for advanced NSCLC [4]. However, patients with NSCLC usually exhibit varying degrees of resistance to radiotherapy, which limits the effectiveness of tumor radiotherapy. Currently, radiotherapy resistance has become one of the important reasons for treatment failure in patients with NSCLC. Therefore, radiosensitizing drugs have become a hot topic in the field of tumor radiotherapy.

Long non-coding RNAs (lncRNAs) are highly conserved transcriptional sequences comprising more than 200 nucleotides, yet they lack the ability to encode proteins [5]. However, lncRNAs play crucial roles in the pathogenesis, progression, and therapeutic response of diverse solid tumors and hematologic malignancies. For example, the high expression of lncRNA CRYBG3 promotes the growth, metastasis (in vitro and in vivo), and radioresistance of NSCLC cells by targeting the eEF1A1/MDM2 pathway [6]. LncRNA CARLo-5 was initially found to be specifically expressed in colon cancer, indicating its association with the invasion, metastasis, and prognosis of colon cancer [7]. LncRNA HOX transcribed antisense RNA (HOTAIR) exerts its inhibitory effect on gene expression through chromatin modification [8]. Recently, we compared the expression of lncRNAs under carbon ion and photon radiation, and found that lncRNA EBLN3P (LNC EBLN3P) was significantly reduced by carbon ion radiation but not X-ray radiation. Furthermore, our findings demonstrated that the downregulation of LNC EBLN3P could increase apoptosis and radiosensitivity in X-ray-irradiated NSCLC cells by regulating the miR-144-3p/TNPO1 axis [9]. Evidence has revealed that mitochondria are the organelles most severely damaged by ionizing radiation (IR), since, except for nuclei, they are the only organelles with their own genome (mtDNA), which is a critical target of ionizing radiation [10]. Thus, mitochondria play a pivotal role in mediating the radiation response by regulating the activity of superoxide dismutases and ROS production in cells. IR-induced alterations in mitochondrial function and ROS production play a crucial role in regulating the evasion of apoptosis and radioresistance, but the molecular mechanisms are still under investigation.

NF-E2-related factor 2 (Nrf2) is a crucial protein that plays a pivotal role in maintaining cellular homeostasis by regulating the expression of genes involved in antioxidant defense and inflammation. It serves as a central regulator, coordinating the cellular response to oxidative stress and other environmental insults, such as ionizing radiation that can damage cells [11]. Under normal circumstances, Nrf2 is sequestered in the cytoplasm by its inhibitory protein Kelch-like ECH-associated protein 1 (Keap1). However, under conditions of oxidative stress or exposure to electrophilic compounds, Nrf2 dissociates from Keap1 and is translocated into the nucleus, where it binds to antioxidant response elements (AREs) located in the promoter regions of target genes [12]. Upon binding to AREs, Nrf2 triggers the transcriptional activation of genes encoding antioxidant enzymes including superoxide dismutase (SOD), catalase, glutathione peroxidase (GPx), heme oxygenase-1 (HO-1), and other ROS scavengers [13]. This process enhances the biosynthesis of these enzymes, which in turn neutralize ROS and protect cellular constituents from oxidative damage. Nrf2 dysregulation has been implicated in various pathological conditions ranging from cancer to neurodegenerative disorders such as Alzheimer’s disease [14]. Therefore, comprehending the mechanistic underpinnings of this transcription factor could pave the way for developing novel therapeutic interventions for these debilitating ailments.

In this study, we investigated the biological mechanism of non-small-cell lung cancer in relation to radiotherapy. Through RNA sequencing (RNA-Seq), we identified LNC EBLN3P as a regulator of lung cancer cell proliferation and radiosensitivity. In addition, it was discovered that LNC EBLN3P can prevent the binding between Keap1 and Nrf2, impeding pan-acidification degradation of Nrf2 and affecting downstream HO-1 expression, which regulates radiation-induced mitochondrial damage. This resulted in a weakened antioxidant capacity within tumor cells and an increased sensitivity to radiation. In conclusion, we propose a potential mechanism by which targeting LNC EBLN3P could lead to the development of novel radiosensitizers.

## 2. Materials and Methods

### 2.1. Cell Culture and Irradiation

A549 and H1299 cells were cultured in RPMI-1640 medium (Sigma, Poole, UK) supplemented with 10% fetal bovine serum (Gibco, Grand Island, NY, USA), 1% penicillin sodium, and 100 µg/mL streptomycin at 37 °C in a humidified incubator containing 5% CO_2_. The cells were irradiated using an RS2000 X-ray machine (Rad Source, Suwanee, GA, USA) at a dose rate of 1.16 Gy/min.

### 2.2. Colony Formation Assay

Colony formation assay was conducted to assess the radiosensitivity of cells. Various cell densities (150, 200, 400, 1000, and 2000 per well) were seeded into six-well plates. After a 24 h incubation period, the cells were exposed to X-ray irradiation at room temperature with doses ranging from 0 to 8 Gy. Following radiation exposure, colonies were allowed to grow for two weeks before being washed twice with PBS and fixed in 70% ethanol solution. Finally, colonies containing more than fifty cells were stained with crystal violet and counted. The colony formation rate for each group (% = (number of colonies/cell number of inoculation) × 100). The survival fraction (SF) for control group was set at 100%, and the SF for treated group (% = (colony formation rate for treated group/colony formation rate for control group) × 100).

### 2.3. Gene Silencing

A549 and H1299 cells were transfected with LNC EBLN3P shRNA lentivirus particles (Sangon, Shanghai, China) for 48 h and then screened with medium containing 2 μg/mL of puromycin (Invitrogen) to obtain stable LNC EBLN3P knockdown cells named as A549/H1299 + shEBLN3P cells. To determine the relationship between LNC EBLN3P and Keap1/Nrf2 pathway, H1299 + shEBLN3P cells were transfected with Keap1 siRNA (Sangon) to silence Keap1.

### 2.4. Western Blot Assay

Total protein was extracted with RIPA buffer and protein concentration was determined using the DC protein Assay Kit I (Bio-Rad, Hercules, CA, USA). All protein samples were denatured at 100 °C for 5 min. An equivalent amount of the protein was isolated using SDS-PAGE and subsequently transferred to a PVDF membrane (Amersham, Arlington, IL, USA). The membrane was immersed in 5% fat-free milk powder, blocked the heterologous antigens at room temperature for 1 h, and then incubated the major antibodies overnight at 4 °C. After washing with PBST three times, membranes were incubated with horseradish peroxidase-conjugated secondary antibody at room temperature for 1 h. Signals were detected and recorded using an ECL kit (Millipore, Bedford, MA, USA) and a polychromatic fluorescence chemiluminescence imaging analysis system (Alpha, San Leandro, CA, USA), and quantified using densitometry (Image J version 1.8.0, National Institutes of Health, Bethesda, MD, USA), respectively. Antibodies used in Western blot included Nrf2, Keap1, GAPDH, MDM2, VEGF, HO-1 (Abcam, Cambridge, MA, USA), p62, and Bcl-2 (Santa Cruz Biotechnology, Piscataway, NJ, USA).

### 2.5. Apoptosis Assay

The cells indicated were collected 48 h after exposure to 4 Gy X-ray irradiation (IR), and apoptotic cells were subsequently analyzed using flow cytometry (BD Biosciences, San Jose, CA, USA) using the Annexin V-PE/7-AAD Apoptosis Detection Kit (Univ, Shanghai, China) according to the manufacturer’s instructions.

### 2.6. Immunofluorescence

Cells were cultured on glass cover slips in a 24-well plate. Two hours after radiation, cells were fixed with 4% paraformaldehyde for 15 min, washed with PBS three times, permeabilized with 0.2% Triton X-100 at room temperature for 15 min, blocked in 5% BSA for an hour, and then incubated overnight at 4 °C with γ-H2AX, Keap1, and Nrf2 antibodies. Following washing thrice with PBST, the slips were incubated with Alexa Fluor 488/559-conjugated goat anti-rabbit secondary antibody for an hour. Subsequently, the samples underwent three rounds of PBST washing and were mounted with DAPI-containing anti-quenching reagent (Beyotime, Shanghai, China) before being imaged using a confocal laser scanning microscope (Olympus, Tokyo, Japan).

### 2.7. Mitochondrial Morphology

After irradiated with 4 Gy X-rays for 24 h, the cells were stained with 100 nM Mitotracker Red (Beyotime, Shanghai, China) in PBS for 40 min. Afterwards, the staining solution was replaced with fresh pre-warmed media and fluorescent images of the cells were captured using a confocal microscope (Nikon, Tokyo, Japan). The size and shape of mitochondria were quantified in a minimum of 50 cells per sample using the Mitochondrial Network Analysis (MiNA) toolset implemented in image j software version 1.8.0, (National Institutes of Health, Bethesda, MD, USA).

### 2.8. Mitochondrial Membrane Potential

The evaluation of mitochondrial membrane potential was conducted using jc-1 (5,5’,6,6’-tetrachloro-1,1’,3,3’-tetraethylbenzimidazolylcarbocyanine iodide). The cells were exposed to a concentration of 5 μg/mL jc-1 for a duration of 20 min at a temperature of 37 °C. subsequently, the cells were rinsed and promptly examined using fluorescence microscopy (Olympus, Tokyo, Japan). for each sample, we acquired five fields of the view and conducted fluorescence image analysis using ImageJ software in three independent experiments.

### 2.9. Mitochondrial Copy Number

Following irradiation for 48 h, mitochondrial copy number was determined using real-time PCR. 12S rRNA encoded by mtDNA and 18S rRNA/GAPDH encoded by nDNA were amplified. The mitochondrial copy number was determined using the comparative threshold cycle (CT) method, which measures the relative quantification of mtDNA/nDNA ratio. For each sample three replicates were analyzed and three independent experiments were performed.

### 2.10. Extracellular Oxygen Consumption

The extracellular oxygen consumption was quantified using the Extracellular Oxygen Consumption Assay kit (Abcam, Cambridge, MA, USA). The irradiated and control cells were seeded in 96-well plates at a density of 5 × 10^5^ cells per well and incubated overnight at 37 °C. Subsequently, the culture media was replaced with 150 µL of fresh media, followed by the addition of 10 μL of extracellular O_2_ consumption dye to each individual well. The wells without O_2_ consumption dye were used as blank controls. Subsequently, all the wells were closed using 100 μL of preheated high-sensitivity mineral oil. Two hours later, extracellular O_2_ consumption signal was measured at Ex/Em wavelengths of 380/650 nm using a fluorescence plate reader pre-set at 37 °C. For each sample, three replicates were analyzed and three independent experiments were performed.

### 2.11. RNA-Seq and GO Analysis

The total RNA of cells was isolated, and ribosomal RNA in samples was removed to maximize the retention of all coding RNA and ncRNA. The obtained RNA was randomly broken into short fragments and used as a template to synthesize the first strand of cDNA using six-base random primers. Next, buffer, dNTPs, RNase H, and DNA polymerase I were added to synthesize the second strand of cDNA. The cDNA was purified using the QiaQuick PCR kit (QIAGEN, Hilden, Germany) and eluted with EB buffer for end repair, and addition of base A and sequencing connectors. The second strand was then degraded using UNG (Uracil N-Glycosylase) enzyme. The fragment size was selected using agarose gel electrophoresis and PCR amplification was performed. The final established sequencing library was sequenced using Illumina HiSeq™ 4000 (Illumina, San Diego, CA, USA). The differentially expressed gene (DEG) analysis was performed on filtered data using the R package “lima”. Genes with an adjusted *p*-value < 0.05 and |fold change| > 2 were selected as significant DEGs. Next, we conducted Gene Ontology (GO) pathway enrichment analysis on DEGs using the R package “clusterProfiler” and presented the results in a bar plot [15].

### 2.12. Statistics

Statistical analysis was performed using GraphPad Prism software version 8.0 (San Diego, CA, USA). All experiments were independently repeated at least three times and all data were represented as means ± standard deviation. Student’s *t*-tests were used for statistical analysis. Differences were considered significant if * *p* < 0.05, ** *p* < 0.01, and *** *p* < 0.001.

## 3. Results

### 3.1. Inhibition of LNC EBLN3P Augments the Radiosensitivity of NSCLC Cells

Our previous studies have reported that the high expression of LNC EBLN3P in NSCLC cells contributes to the regulation of radiosensitivity. To further clarify the role of LNC EBLN3P in conferring radioresistance to NSCLC cells, we employed lentiviral vectors expressing short hairpin RNAs (shRNAs) to knockdown its expression in A549 and H1299 cells. The interference efficiency of two shRNAs was assessed at 48 h post-transfection. As depicted in Figure 1A,B, shEBLN3P#2 exhibited the most optimal knockdown efficacy and was therefore selected for further investigation. Moreover, the impact of LNC EBLN3P knockdown on the colony-forming ability was evaluated in A549 and H1299 cells subjected to varying doses of irradiation. A significant reduction in the number of colonies formed by A549 and H1299 cells transfected with LNC EBLN3P-shRNA was observed compared to control cells (Figure 1C–F). The phosphorylated form of γH2AX has been validated as a sensitive biomarker for DNA double-strand breaks (DSBs). To evaluate the impact of LNC EBLN3P inhibition on radiation-induced DNA DSBs, the number of γ-H2AX foci was quantified using immunofluorescence microscopy (Figure 1G–I). Moreover, the incidence of radiation-induced apoptosis was significantly elevated in A549 and H1299 cells following LNC EBLN3P knockdown (Figure 1J–L). The collective findings demonstrate that the downregulation of LNC EBLN3P significantly enhances the radiosensitivity of NSCLC cells.

### 3.2. Inhibition of LNC EBLN3P Increases ROS Production and Mitochondrial Injury in NSCLC Cells

The radiation-induced generation of ROS can lead to intracellular oxidative stress, resulting in irreversible cellular damage. Therefore, intracellular ROS levels were monitored in H1299 cells at 1 h after exposure to 4 Gy of X-ray irradiation. The downregulation of LNC EBLN3P was found to effectively decrease radiation-induced ROS generation in H1299 cells, as demonstrated by Figure 2A,B. Numerous studies have demonstrated that excessive intracellular production of ROS induced by radiation is associated with enhanced mitochondrial damage, ultimately resulting in lethal cellular injury and death. To investigate the role of LNC EBLN3P in radiation-induced mitochondrial damage, we assessed mitochondrial length and membrane potential in H1299 cells with LNC EBLN3P knockdown following exposure to 4 Gy X-ray irradiation. Mitochondrial fluorescence staining suggested that H1299 + shEBLN3P cells displayed more severe mitochondrial fragmentation, characterized by small, round fragments, than H1299 + shNC cells (Figure 2C). Consistently, the measurement of mitochondrial length showed a significant decrease in average mitochondrial length in H1299 + shEBLN3P cells (Figure 2D). Mitochondrial dynamics are regulated by fission and fusion processes, which are mediated by specific proteins such as dynamin-related protein (Drp1), mitofusin1, and mitofusin2 (Mfn1/2). After exposure to 4 Gy X-rays, cells with LNC EBLN3P knockdown showed a higher expression of the fission gene Drp1 and a lower expression of the fusion gene MFN1 compared to the control cells (Figure 2E and Appendix A).

The mitochondrial DNA copy number, as an indicator of mitochondrial damage, is expressed by the ratio of 12S rRNA/GAPDH. Compared to H1299 + shNC cells, a significant reduction in mitochondrial copy number was observed in H1299 + shEBLN3P cells (Figure 2F). In addition to assessing mitochondrial DNA alterations, changes in mitochondrial functions were also evaluated, including oxygen consumption and membrane potential. As shown in Figure 2G, cells with downregulated LNC EBLN3P exhibited a decrease in overall oxygen consumption compared to control cells at the 2 h post-irradiation time point. JC-1 staining was used to detect the mitochondrial membrane potential, with red fluorescence indicating an increase in mitochondrial membrane potential and green indicating a decrease in mitochondrial membrane potential. As shown in Figure 2H,I, the red fluorescence was more intense in unirradiated H1299 + shNC and H1299 + shEBLN3P cells, indicating a higher mitochondrial membrane potential under normal conditions. After exposure to 4 Gy X-rays, weaker red fluorescence and stronger green fluorescence were observed in H1299 + shEBLN3P cells compared to to H1299 + shNC cells, suggesting more pronounced mitochondrial damage occurred in the former after radiation. Additionally, we conducted an antioxidant rescue experiment and found that the administration of 2 mM N-acetylcysteine (NAC) effectively mitigated the elevation of ROS and meliorated mitochondrial damage and apoptosis induced by the knockdown of LNC ENLN3B. The knockdown of LNC ENLN3B resulted in a reduction in PGC-1α levels in irradiated cells, which was subsequently restored by the addition of NAC (Appendix A). In summary, the knockdown of LNC EBLN3P could exacerbate radiation-induced mitochondrial morphological damage and dysfunction.

### 3.3. LNC EBLN3P Targets the Keap1-Nrf2 System

To explore the genes regulated by LNC EBLN3P, we performed RNA sequencing for shNC- and shEBLN3P-transfected cells and performed GO pathway enrichment analysis for the sequencing results. As shown in Figure 3A, the antioxidant activity of cells was obviously inhibited in the shEBLN3P group. Given Keap1-Nrf2 system plays an essential role in antioxidant reaction, we detected the transcriptional expression of Nrf2 and Keap1, and found that the silencing of LNC EBLN3P did not affect the mRNA expression of these two genes (Figure 3B,C). However, the silencing of LNC EBLN3P significantly inhibited the Nrf2 protein expression but had no effect on Keap1 protein expression (Figure 3D–F and Appendix A). According to these data, we hypothesized that LNC EBLN3P may play a protective role to keep Nrf2 from being degraded by Keap1.

### 3.4. LNC EBLN3P Interacts with Nrf2 to Inhibit the Association between Keap1 and Nrf2

To further investigate the role of LNC EBLN3P in the Keap1/Nrf2 pathway, we conducted RNA pull down and RNA immunoprecipitation (RIP) assay, which revealed a direct interaction between Nrf2 and LNC EBLN3P (Figure 4A,B and Appendix A). Furthermore, immunoprecipitation (IP) analysis using the Keap1 protein antibody revealed that the interaction between Keap1 and Nrf2 was potentiated by LNC EBLN3P downregulation (Figure 4C and Appendix A). It has been reported that Keap1 and Nrf2 interact in the cytoplasm, leading to the ubiquitination-mediated degradation of Nrf2, thereby reducing its nuclear translocation and regulating downstream gene expression. Both Western blot and immunofluorescence tests showed a significant reduction in Nrf2 in the nucleus after LNC EBLN3P knockdown. Additionally, our findings indicated that radiation can increase the expression and phosphorylation of Nrf2, as well as promote its translocation to the nucleus. However, knockdown of LNC EBLN3P resulted in decreased levels of Nrf2 in both the cytoplasm and nucleus (Figure 4D–G, Appendix A). These findings suggested that downregulation of LNC EBLN3P can attenuate the expression and transcriptional activity of Nrf2, consequently diminishing the antioxidative stress capacity of lung cancer cells.

### 3.5. LNC EBLN3P Modulates the Radiosensitivity of NSCLC Cells by Targeting the Keap1/Nrf2/HO-1 Signaling Pathway

Therefore, the overexpression of LNC EBLN3P in NSCLC cells is thought to impede the interaction between Nrf2 and Keap1, leading to the stabilization of Nrf2 and subsequent transcriptional activation of its target genes. High Nrf2 activity in cancer cells leads to the upregulation of various proteins associated with cancer progression, including ABCG2, VEGF, HO-1, Bcl-2, p62, and MDM2. Similarly, we observed that the knockdown of LNC EBLN3P led to changes in the expression of Nrf2 target genes. Specifically, there was a significant decrease in the transcriptional expression levels of VEGF, Bcl-2, HO-1, and p62. However, only HO-1 continued to decrease after irradiation (Figure 5A–C and Appendix A). After knocking down LNC EBLN3P in H1299 cells, the binding between Keap1 and Nrf2 was increased, and this promoted Nrf2 degradation and subsequently affected the expression of HO-1. Therefore, we proceeded to knockdown Keap1 in H1299 + shEBLN3P cells as a means of confirming the impact of LNC EBLN3P on Nrf2 function via Keap1. As shown in Figure 5D and Appendix A, the knockdown of Keap1 prevented the reduction in Nrf2 and HO-1 levels induced by the downregulation of LNC EBLN3P in H1299 + shEBLN3P cells, whether treated with radiation or not. Similarly, Keap1 silencing alleviated the increase in ROS production (Figure 5E,F), mitochondrial damage (Figure 5G–K) and the enhancement in radiosensitivity (Figure 5L,M) induced by the downregulation of LNC EBLN3P in irradiated H1299 + shEBLN3P cells.

## 4. Discussion

Although significant advances have been made in the field of medicine targeting NSCLC, the prognosis and survival rates for NSCLC remain disappointing, with a high mortality rate due in part to tumor tolerance to chemoradiotherapy [16,17]. For unresectable NSCLC, the combination of chemotherapy and radiotherapy can suppress distant metastasis of the tumor and improve the survival rate of patients. However, the effectiveness of radiotherapy is limited by the radioresistance of lung cancer cells [18]. LncRNAs play a variety of biological functions by regulating gene expression and function at the transcriptional, and translational and post-translational levels [19]. It is estimated that there are more than 100,000 lncRNAs in humans, and their number is still increasing rapidly [20]. So far, only a very small number of lncRNA functions have been annotated [6,7], and various methods have been developed to explore the expression, distribution, and function of lncRNAs [19]. LncRNAs may serve as tumor promotors or inhibitors by modulating the growth and development of a variety of cancer types, such as NSCLC [21]. Recent studies have shown that lncRNA plays an important role in regulating the radiosensitivity of tumor cells and has the potential to be a novel biological target for NSCLC treatment [20,22,23]. As an important lncRNA, LNC EBLN3P is upregulated and promotes tumor progression in a variety of tumors. Studies have shown that LNC EBLN3P can promote the progression of osteosarcoma and rectal cancer by regulating different ceRNA axes, and can lead to the resistance of osteosarcoma cells to methotrexate [24,25,26]. The downregulation of LNC EBLN3P using various methods (such as siRNA, shRNA silencing pathway, or carbon ion beam irradiation) can enhance the therapeutic effect by inhibiting malignant behavior of tumor cells, inducing apoptosis of tumor cells, and increasing radiation sensitivity [9,27,28]. Therefore, LNC EBLN3P is a carcinogenic factor and therapeutic target for tumors. This study revealed that LNC EBLN3P increases the radiosensitivity of NSCLC cells by regulating the Nrf2/HO-1 axis and mitochondrial oxidative stress damage.

The biological basis of tumor radiotherapy is that ionizing radiation directly destroys biological macromolecules or ionizes intracellular water molecules to produce a large number of primary radiolysis products of water molecules, including ·OH and H_2_O_2_ in ROS, which then attack nucleic acid molecules, biomembranes, and enzyme molecules. It leads to the breakdown of DNA molecules, an imbalance of membrane lipid peroxidation, metabolism, and signal transduction, which leads to apoptosis, necrosis, autophagy-dependent cell death, immunogenic cell death, *etc*. [29,30,31]. In fact, a large number of ROS produced in cells attack DNA molecules, leading to a series of damage such as DNA base damage, crosslinking, single-strand break (SSB) and double-strand break (DSB) [32,33]. The most important damage is a DSB, which is a fatal blow to cells, and DSB leads to the phosphorylation of histone H2AX. H2AX ser139 phosphorylation is abundant and rapid, and has a good correlation with DSB, so it is often used to detect DNA damage [34,35].

Mitochondria are small double-membraned organelles in the cytoplasm of eukaryotic cells that function as the main oxygen-consuming powerplants, and are the main source of physiological ROS levels [36,37,38]. In the mitochondrial ETC and OXPHOS system, although mitochondria are equipped with an outer and an inner membrane for protection, from 1% to 5% of electrons may leak out during OXPHOS, and further reactions with oxygen cause the formation of ROS [39]. The above processes are damaged by IR-mediated oxidative stress, which interferes with mitochondrial structure and function and strongly increases the level of ROS produced, forming a vicious cycle [40,41,42]. Excessive ROS production will lead to mitochondrial dysfunction [43]. ROS directly generated by radiation and derived from mitochondria act on and destroy organelle membranes, such as plasma membrane or mitochondrial membrane, resulting in changes in intracellular ions, potential, and osmotic pressure [29]. These phenomena can be detected using the Bcl-2 family proteins located on the mitochondrial membrane, thus activating the intrinsic apoptosis pathway and forming the Bax–Bak pore on the mitochondrial membrane. When cytochrome c and other pro-apoptotic factors are released into the cytoplasm, caspase-9 and its downstream effector procaspases-3 are activated and the final execution pathway of apoptosis is initiated [44,45]. Due to their high metabolic and proliferative activity, cancer cells generally exhibit increased mitochondrial mass and ROS production, and cancer cell mitochondria may be more sensitive than normal cell mitochondria to additional damage to IR and ROS induction, making them susceptible to perturbations of cell metabolism and ROS produced by external genotoxic substances such as IR. This exacerbates cellular oxidative stress and causes long-term reactions such as cell death, inflammation, and immunogenic exposure [46]. Therefore, radiation-induced mitochondrial oxidative stress injury is an important factor promoting cell radiation sensitivity.

Many studies have pointed out that lncRNA can also affect the radiosensitivity of tumor cells by regulating the balance between apoptotic proteins and anti-apoptotic proteins, autophagy, and the expression level of target proteins related to epithelial mesenchymal transformation through the ceRNA mechanism [22,47,48,49]. In our work, the number of γ-H2AX foci in tumor cells and the ratio of apoptotic cells rose following 4 Gy X-ray irradiation, leading to an increase in the cell’s radiosensitivity. The suppression of the LNC EBLN3P gene furthered the aforementioned event. We examined the related indicators of ROS levels in tumor cells and mitochondrial damage in order to investigate the causes of the increased radiosensitivity of tumor cells after LNC EBLN3P knockdown. We discovered that LNC EBLN3P knockdown enhanced the increase in ROS levels in tumor cells and the structural damage and functional inhibition caused by 4 Gy X-ray irradiation. Therefore, the downregulation of LNC EBLN3P plays a radiosensitizing role by increasing the radiation-induced mitochondrial oxidative stress damage in tumor cells.

Nuclear factor (erythroid-derived 2)-like factor 2 (NFE2L2), also commonly known as NF-E2-related factor 2 (Nrf2), first appeared in the scientific literature in 1994 [50]. Since then, an extensive body of research has unveiled the crucial role played by Nrf2 in orchestrating the cytoprotective effects associated with redox homeostasis maintenance, and the regulation of antioxidant responses [51]. The Keap1/Nrf2 pathway, consisting of Keap1 and Nrf2, stands as a critical molecular axis governing antioxidation mechanisms [52]. In instances devoid of oxidative stress, Keap1 effectively mediates the degradation of Nrf2, ensuring the preservation of cellular equilibrium [53]. In contrast, the occurrence of oxidative stress conditions instigates the modification of reactive cysteine residues, culminating in translocation of Nrf2 to the nucleus once situated in the nucleus, Nrf2 diligently binds to antioxidant response elements, thereby initiating the expression of an assorted array of cytoprotective enzymes [54]. The activation of Nrf2 presents a dual-edged phenomenon, encompassing both advantageous and disadvantageous consequences. On the one hand, Nrf2 activation diligently upholds normal cellular homeostasis, while on the other, it has been implicated in fueling the rapid growth, proliferation, and radiation insensitivity commonly observed in cancer cells [55]. Notably, heme oxygenase-1 (HO-1), one of the three distinct isoforms belonging to the heme oxygenase (HO) family, has emerged as a pivotal component in Nrf2-mediated mechanisms underlying tumor resistance to anticancer therapies [56,57].

In the current research, we have embarked upon a pioneering endeavor to unravel the intricate role played by LNC EBLN3P in the regulation of the Nrf2/HO-1 axis within the realm of cancer therapy. Our comprehensive findings have revealed a momentous discovery: the knockdown of LNC EBLN3P engenders a conspicuous enhancement in radiosensitivity. Intriguingly, our study has unveiled a noteworthy decline in Nrf2 expression subsequent to LNC EBLN3P knockdown, while exhibiting no substantial alterations in the expression of Keap1. Moreover, a meticulous assessment comparing the levels of Nrf2 within the cytoplasm and nucleus has indicated a declining trend in the cytoplasmic compartment juxtaposed with an escalating trend in the nuclear compartment. This intriguing pattern strongly suggests a reduced rate of Nrf2 degradation and an amplified influx of Nrf2 into the nucleus. In light of these observations, we have elucidated that LNC EBLN3P exerts its influence by selectively binding to Nrf2, thereby impeding its degradation orchestrated by Keap1 and facilitating its translocation into the nucleus. The antioxidant protein HO-1 is regulated by Nrf2 as its main target protein and increases with Nrf2 [56]. Consistent with our expectations, the knockdown of LNC EBLN3P has manifested in a significant reduction in the expression levels of Nrf2 and HO-1, corroborating the tightly regulated relationship between LNC EBLN3P, Nrf2, and HO-1. Collectively, our findings unequivocally suggest that the silencing of LNC EBLN3P diminishes its binding affinity to Nrf2, thereby facilitating an augmented interaction between Nrf2 and Keap1. Consequently, this promotes the degradation of Nrf2, attenuates its translocation into the nucleus, and ultimately culminates in a substantial decrease in HO-1 expression. Thus, these cascading events culminate in the attenuation of cellular antioxidant capacity and a marked escalation in radiosensitivity.

Contemporary research has shed light on the remarkable involvement of lncRNAs in a myriad of intricate cellular regulatory processes, thus positioning them as valuable biomarkers for discerning distinct subgroups and prognostic determinants [58,59]. LNC EBLN3P has emerged as a focal point of interest, given its dysregulation in diverse malignancies such as osteosarcoma, liver cancer, and breast cancer [24,60,61]. Furthermore, in a previous study by our team, the expression of LNC EBLN3P in lung cancer cell lines was much higher than in the normal lung bronchial epithelial cell line BEAS-2B, indicating its potential as a therapeutic target in NSCLC [9]. Nevertheless, the precise targeting and inhibition strategies for LNC EBLN3P in the context of tumor eradication remain an uncharted territory, thus signifying an imperative avenue for our forthcoming investigations. The optimal dosage and fractionation scheme for radiation therapy in cancer patients present a realm of uncertainty, necessitating meticulous tailoring to individual clinical scenarios. Regrettably, our study was subject to certain limitations, as we did not explore the impact of varying dose–time fractions of radiotherapy on the LNC EBLN3P-Nrf2/HO-1 axis. This represents an unresolved issue that warrants future exploration, wherein rigorous experimentation can pave the way for the identification of the most efficacious dose–time fractions. At present, oxygen mimetics have emerged as vital elements in the sensitization of radiation therapy, with nitro compounds and nitric oxide standing as prime examples [62]. The combination of LNC EBLN3P-targeted inhibitors and oxygenimetic agents provides superior efficacy, while the combination with specific radioprotective agents can protect against severe damage to normal tissue. Undoubtedly, the exploration of LNC EBLN3P and its intricate interactions within the Nrf2/HO-1 axis opens up a rich landscape of research possibilities, both in terms of therapeutic interventions and the optimization of radiation therapy. Further investigations hold the potential to unravel the hidden intricacies underlying these phenomena and pave the way for novel strategies in the quest for effective tumor management.

## 5. Conclusions

In conclusion, we report that the inhibition of LNC EBLN3P can enhance the radiosensitivity of NSCLC cells by promoting radiation-induced mitochondrial damage through modulation of the Keap1/Nrf2/HO-1 axis. Therefore, siRNAs targeting LNC EBLN3P are anticipated to emerge as novel radiosensitizers for the radiotherapy of NSCLC.

## Figures and Tables

**Figure 1 biology-12-01208-f001:**
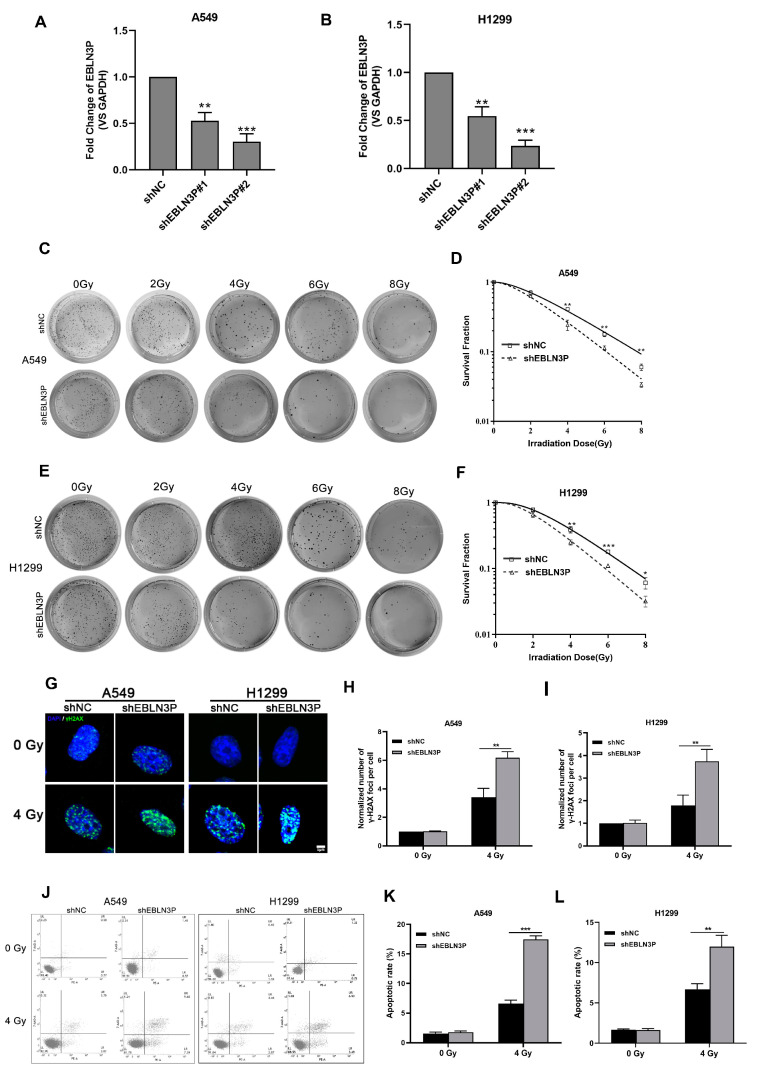
Inhibition of LNC EBLN3P augments the radiosensitivity of NSCLC cells. (**A**,**B**) The interference efficiency of two shRNAs was assessed at 48 h post-transfection in A549 and H1299 cells. (**C**–**F**) Colony formation and survival curves of A549 and H1299 cells transfected with shEBLN3P or shNC were measured after X-ray irradiation at different doses (0, 2, 4, 6, and 8 Gy). (**G**) Representative images of γ−H2AX foci (green) formation in A549 and H1299 cells transfected with shEBLN3P or shNC at 2 h after 4 Gy X-ray irradiation. Nuclei were stained with DAPI (blue). Scale bar = 5 μm. (**H**,**I**) Quantitation of γ−H2AX foci formation in shEBLN3P or shNC-transfected cells. The foci number in each group was normalized to that in shNC cells of 0 Gy group. (**J**–**L**) Apoptotic cells were analyzed using flow cytometry at 48 h after 4 Gy X-ray irradiation. Data are represented as means ± SD (three biological replicates; * *p* <0.05, ** *p* < 0.01, *** *p* < 0.001).

**Figure 2 biology-12-01208-f002:**
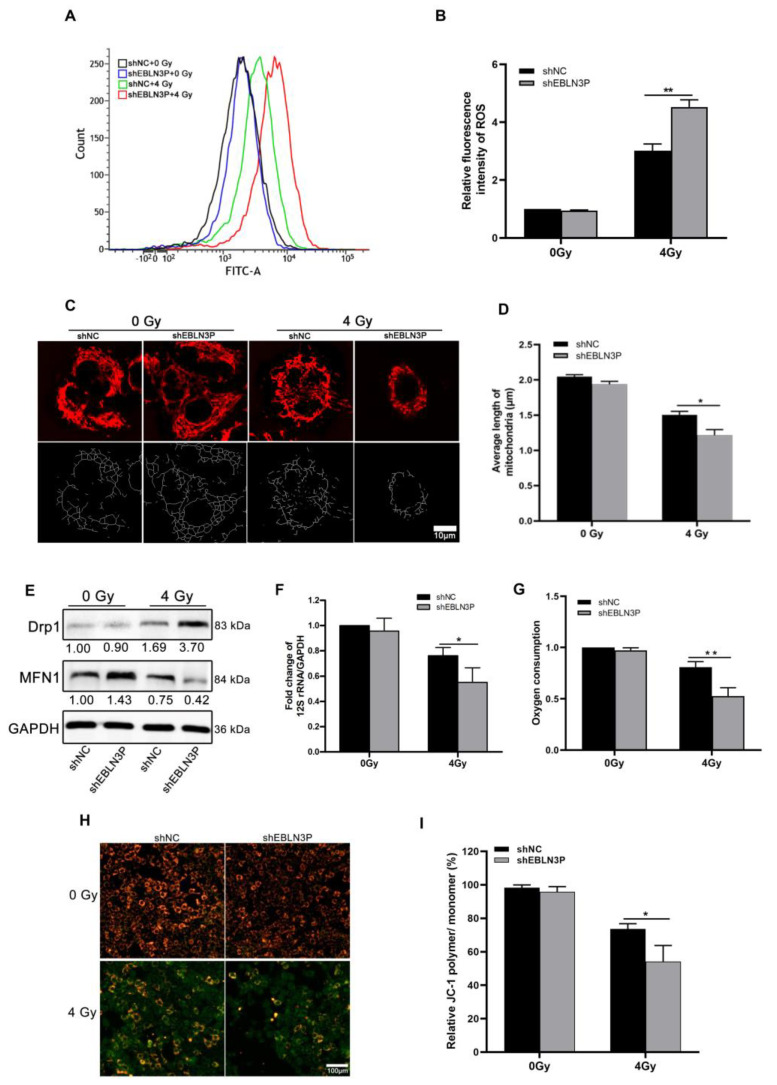
Inhibition of LNC EBLN3P increases ROS production and mitochondrial injury in NSCLC cells. (**A**,**B**) Intracellular ROS levels were monitored in H1299 cells transfected with shEBLN3P or shNC at 1 h after exposure to 4 Gy of X−ray irradiation. (**C**,**D**) Representative mitochondrial morphology in H1299 cells was visualized using Mitotracker Red staining (**top row**), and image analysis was performed on the “skeletonized” conversion (**bottom row**). Scale bar = 10 μm. (**E**) Western blot analyses of proteins related to mitochondrial dynamics at 24 h after 4 Gy X-ray irradiation. Relative densitometry values for the representative blots are given below each band. (**F**) Fold change of mitochondrial copy number determined using real-time PCR of 12S rRNA/GAPDH. (**G**) Oxygen consumption. (**H**,**I**) Quantification of JC−1 staining (red/green fluorescence) for the detection of mitochondrial membrane potential (MMP). JC−1 polymers show red and JC−1 monomers green fluorescence. Scale bar = 100 μm. Data are represented as means ± SD (three biological replicates; * *p* <0.05, ** *p* < 0.01).

**Figure 3 biology-12-01208-f003:**
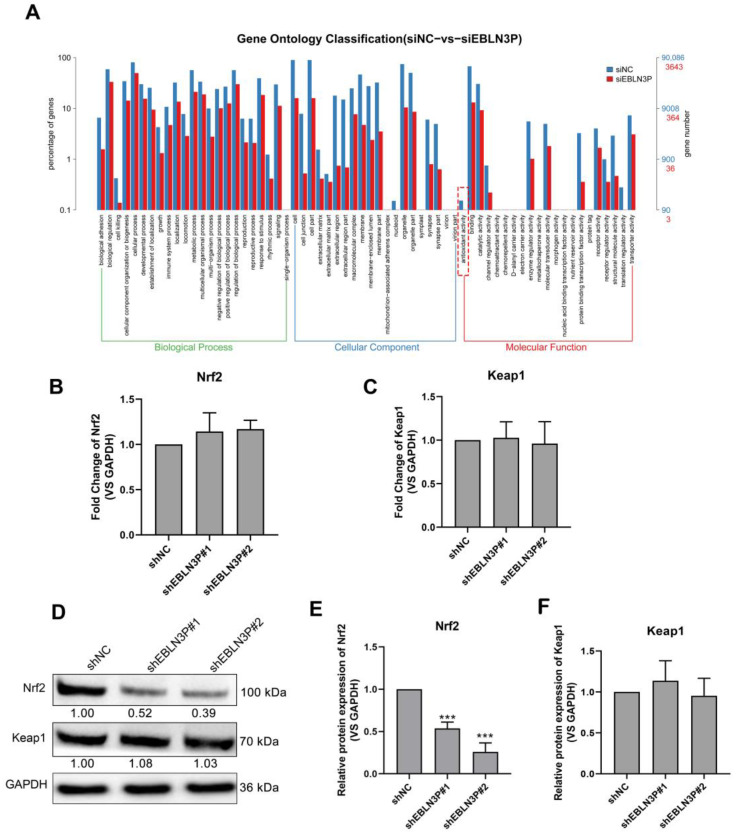
LNC EBLN3P targets the Keap1-Nrf2 system. (**A**) GO pathway enrichment analysis. The dashed box highlights the antioxidant activity inhibited by LNC EBLN3P knockdown. (**B**,**C**) qPCR analyses of the transcription expression of Nrf2 and Keap1 in H1299 cells transfected with shNC or shEBLN3P. Relative densitometry values for the representative blots are given below each band. (**D**–**F**) Western blot analyses of Nrf2 and Keap1. Data are represented as means ± SD (three biological replicates; *** *p* < 0.001).

**Figure 4 biology-12-01208-f004:**
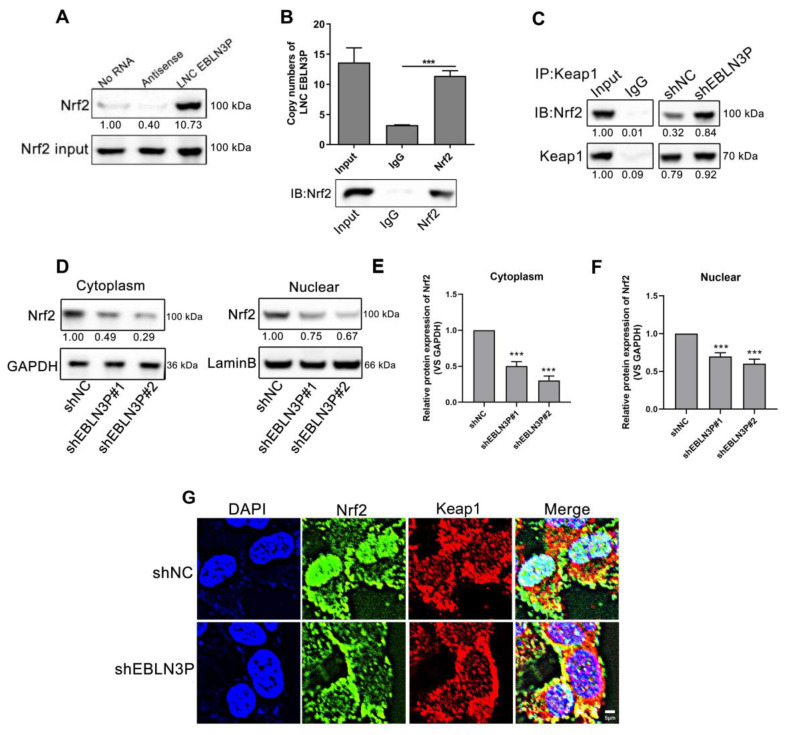
LNC EBLN3P interacts with Nrf2 to inhibit the association between Keap1 and Nrf2. (**A**) RNA pull down analysis was used to determine the Nrf2-LNC EBLN3P interaction in H1299 cells. (**B**) Histogram of LNC EBLN3P enrichment after RNA immunoprecipitation (IP) assays. Nrf2 antibodies were used (three biological replicates; *** *p* < 0.001). (**C**) Immunoprecipitation followed by Western blot was used to detect the interaction of Nrf2 with Keap1. All experiments were independently repeated at least three times. (**D**–**F**) The expression of Nrf2 was assessed using Western blot analysis in both the nucleus and cytoplasm. Relative densitometry values for the representative blots are given below each band. Data are represented as means ± SD (three biological replicates; ** *p* < 0.01, *** *p* < 0.001). (**G**) The co-localization of Nrf2 (green) and Keap1 (red) in cells was detected through an immunofluorescence assay. Nuclei were stained with DAPI (blue). Scale bar = 5 μm.

**Figure 5 biology-12-01208-f005:**
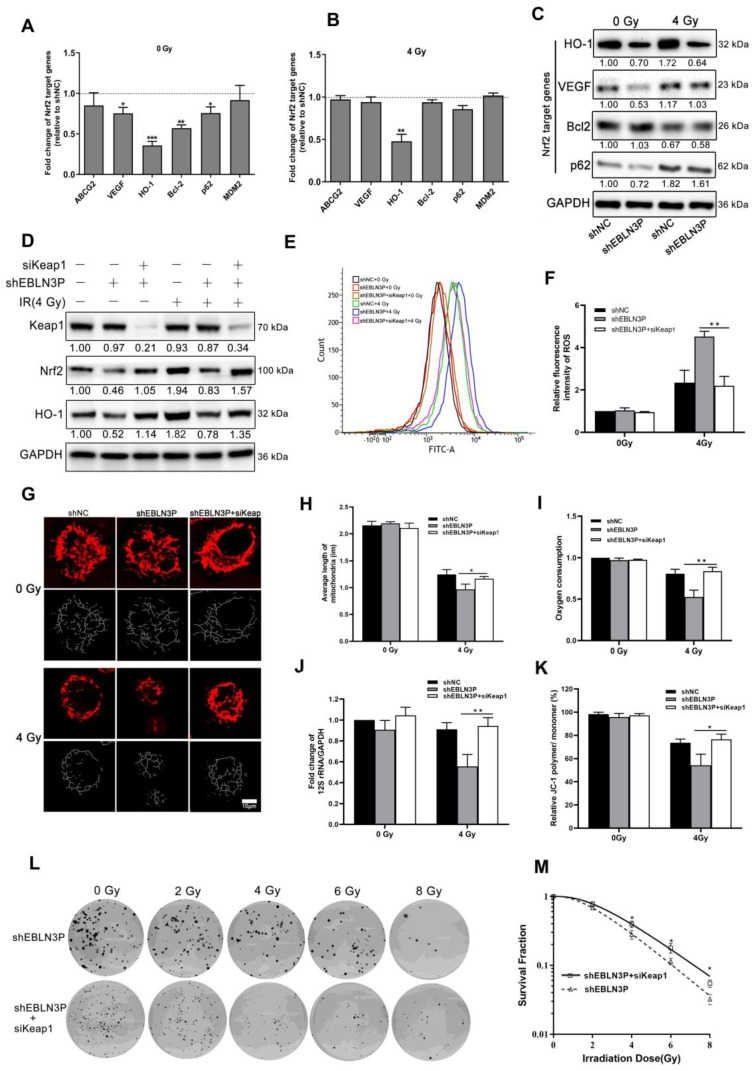
LNC EBLN3P modulates radiation-induced mitochondrial damage in lung cancer cells by targeting Keap1/Nrf2/HO−1 axis. (**A**,**B**) Fold change in Nrf2 target genes in H1299 + shEBLN3P cells, with or without irradiation, compared to H1299 + shNC cells. (**C**) Nrf2 target gene expression was analyzed using Western blot. (**D**) The expression of Nrf2 and HO−1 was assessed in H1299 + shEBLN3P/shNC cells treated with siKeap1. Relative densitometry values for the representative blots are given below each band. (**E**,**F**) ROS production was tested using flow cytometry in H1299 + shEBLN3P/shNC cells treated with siKeap1, with or without irradiation. (**G**–**K**) Alterations in both the structure and function of mitochondria. Scale bar = 10 μm. (**L**,**M**) Colony formation and survival curves were measured for H1299 + shEBLN3P/shNC cells treated with siKeap1 after X-ray irradiation at different doses (0, 2, 4, 6, and 8 Gy). Data are represented as means ± SD (three biological replicates; * *p* < 0.05, ** *p* < 0.01, *** *p* < 0.001).

## Data Availability

The data presented in this study are available in this article.

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
