# Peer review of "Inhibition of LNC EBLN3P Enhances Radiation-Induced Mitochondrial Damage in Lung Cancer Cells by Targeting the Keap1/Nrf2/HO-1 Axis"

_biology, 2023, doi:10.3390/biology12091208_

Round 1

Reviewer 1 Report

            The authors revealed that lncRNA EBLN3P enhances radiosensitivity of non-small cell lung cancer (NSCLC) cell lines A549 and H1299 via downregulation of Nrf2 i.e., ROS increasing in the cell lines. In this manuscript, although the authors aiming to overcome the radioresistance of NSCLC cells, regular cell lines were used. You can acquire radioresistant cancer cells repeated irradiation or extracting the cancer stem cell fraction. Besides, radioresistant cell lines generally acquired increasing ROS scavenging capability, so it is unclear that LNC EBLN3P enhances the radiosensitivity of radioresistant cell lines.

Does LNC EBLN3P directly regulate the expression of Nrf2? You can confirm whether the base sequences are complementary or not.

Is the copy number of LNC EBLN3P increasing in A549 and H1299 cell lines compared with normal lung cell lines? Please discuss whether it can be a target for cancer therapy in terms of comparison with normal cells.

Author Response

Dear reviewer,

Thank you for your kind comments on our manuscript (biology-2506894) entitled “Inhibition of LNC EBLN3P enhances radiation-induced mitochondrial damage in lung cancer cells by targeting the Keap1/Nrf2/HO-1 axis”. We appreciate the time and effort that you dedicated to providing feedback on our manuscript and are grateful for your insightful comments on our paper. According to your comments and suggestions, we have revised the manuscript carefully and made extensive modifications to the original manuscript. Our responses are given in a point-by-point manner as listed below.

Response to Reviewer 1 Comments

  1. The authors revealed that lncRNA EBLN3P enhances radiosensitivity of non-small cell lung cancer (NSCLC) cell lines A549 and H1299 via downregulation of Nrf2 i.e., ROS increasing in the cell lines. In this manuscript, although the authors aiming to overcome the radioresistance of NSCLC cells, regular cell lines were used. You can acquire radioresistant cancer cells repeated irradiation or extracting the cancer stem cell fraction. Besides, radioresistant cell lines generally acquired increasing ROS scavenging capability, so it is unclear that LNC EBLN3P enhances the radiosensitivity of radioresistant cell lines.

We really appreciate your constructive suggestion. Actually, the radioresistant lung cancer cells indeed possess relatively higher expression of Nrf2 as well as the increasing ROS scavenging capability as demonstrated in some related literatures (Int J Mol Sci, 2016, 17, 997; Free Radical Bio Med, 2012, 53: 807-816). Besides, it has been suggested that lung adenocarcinoma is not highly sensitive to radiation (Int J Biol Sci, 2020, 16(6): 1010-1022), thus the two lung adenocarcinoma cell lines (A549 and H1299) were chosen to study the radiosensitizing effect of LNC EBLN3P inhibition in this work. Moreover, Liu et al established a radioresistant cell line named A549R by interval irradiating A549 cells with a total of 20 Gy and found that Nrf2 inhibition radiosensitized both A549 and A549R cells significantly (Biochem Pharmacol, 2022, 199: 114981). In brief, it can be found from both our results and the published papers that Nrf2 inhibition could enhance the radiosensitivity of radioresistant lung adenocarcinoma cells. Hopefully, it is expected to verify the effect of Nrf2 inhibition caused by LNC EBLN3P knockdown on established radioresistant lung cancer cells in our future research.

  1. Does LNC EBLN3P directly regulate the expression of Nrf2? You can confirm whether the base sequences are complementary or not.

Thank you so much for your suggestion and reminder. We have checked the interaction between LNC EBLN3P and Nrf2 mRNA using an online tool “LncRRIsearch” developed by Fukunaga et al (Frontiers in Genetics, 2019, 10: 462). However, no interaction was identified.  

  1. Is the copy number of LNC EBLN3P increasing in A549 and H1299 cell lines compared with normal lung cell lines? Please discuss whether it can be a target for cancer therapy in terms of comparison with normal cells.

Thank you for your constructive suggestion. The relative expression of LNC EBLN3P in A549 and H1299 cell lines is much higher than that in BEAS-2B (a normal lung bronchial epithelial cell line) as determined in Figure 2A of the paper we previously published in Cancers (Cancers 2023, 15, 511), which was also cited in this paper. Considering the higher expression in lung cancer cells, LNC EBLN3P may be a good target for the development of new cancer therapy, which has also been supplemented in the last paragraph of Discussion section.

Reviewer 2 Report

Tang et al has shown interesting results on the role of LNC EBLN3P in regulating mitochondrial function and oxidative stress in lung cancer cells. Though the experimental design is adequate but there are some notable concerns in the study.

1. Lung cancer cells or any cancer cells have intrinsically high levels of ROS, to counteract with the harsh extracellular environments. Biologically relevant dosages of radiation must be used. 4 Gy is quite high it seems, what is the rational of using this dose? The authors should consider using a dose dependent study to show the effect on mitochondria and oxidative stress with or without knockdown of LNC EBLN3P.

2. Mitochondrial function was diminished in radiation-treated conditions, what is the status of mitophagy and mitochondrial metabolism in these cells? What is triggering this mitochondrial abnormality? Is there any changes in PGC-1alpha levels? A rescue experiment with NAC or other anti-oxidant to show that oxidative stress is the reason for mitochondrial abnormality must be performed. The link between radiation induction-oxidative stress-mitochondrial dysfunction is missing.

3. Nrf2 is a transcription factor, hence just the total level of Nrf2 do not justify its nuclear translocation status. Either nuclear-cytosolic fractionation or WB with appropriate phosphorylation antibody must be performed. Oxidative stress mediators and Nrf2 targets like SOD, NQO1, CAT enzyme levels and activities must be performed.

4. The regulation of Nrf2-Keap1 is very naive as presented by the authors. There are other factors that regulate the entire process including p62, p21 etc. In-depth analysis must be done to show binding of these proteins with Nrf2 under different experimental conditions to prove the signaling pathway.

5. Appropriate negative and positive controls must be included to show that oxidative stress as induced by the radiation is key in triggering the underlying  changes. Also, apoptotic mediators like Bax, Bcl2, cytochrome C and caspase3 must be estimated by WB.

Appropriate.

Reviewer 3 Report

The manuscript titled “Inhibition of LNC EBLN3P enhances radiation-induced mitochondrial damage in lung cancer cells by targeting the Keap1/Nrf2/HO-1 axis” by Tang et al. sheds light on the complex biology of radiotherapy resistance in NSCLC.  Lung cancer is the leading cause of cancer-related deaths worldwide, with a pressing challenge being radiotherapy resistance in NSCLC. To address this, authors have investigated NSCLS’s molecular mechanisms in relation to radiotherapy and identified LNC EBLN3P as a regulator of lung cancer cell proliferation and radiosensitivity. LNC EBLN3P was found to interact with Nrf2, inhibiting the Keap1 and Nrf2 system’s association. Inhibition of LNC EBLN3P resulted in increased ROS production and mitochondrial injury in NSCLC cells. These exciting findings suggest that targeting LNC EBLN3P could offer a promising avenue for developing novel radiosensitizers to combat radiotherapy resistance in NSCLC.

However, some improvements below are suggested.

RNA-seq and GO pathway enrichment analyses were done to explore the genes regulated by EBLN3P. If RNA-seq was done as mentioned on the manuscript, then please provide the methods of its preparation and data analysis. Also, resolution of Fig 3A could be improved.

LNC EBLN3P and lncRNA EBLN3P are both are being used interchangeably. Shouldn’t we follow one format for the consistency?

Author Response

Dear reviewer,

Thank you for your kind comments on our manuscript (biology-2506894) entitled “Inhibition of LNC EBLN3P enhances radiation-induced mitochondrial damage in lung cancer cells by targeting the Keap1/Nrf2/HO-1 axis”. We appreciate the time and effort that you dedicated to providing feedback on our manuscript and are grateful for your insightful comments on our paper. According to your comments and suggestions, we have revised the manuscript carefully and made extensive modifications to the original manuscript. Our responses are given in a point-by-point manner as listed below.

Response to Reviewer 3 Comments

The manuscript titled “Inhibition of LNC EBLN3P enhances radiation-induced mitochondrial damage in lung cancer cells by targeting the Keap1/Nrf2/HO-1 axis” by Tang et al. sheds light on the complex biology of radiotherapy resistance in NSCLC.  Lung cancer is the leading cause of cancer-related deaths worldwide, with a pressing challenge being radiotherapy resistance in NSCLC. To address this, authors have investigated NSCLS’s molecular mechanisms in relation to radiotherapy and identified LNC EBLN3P as a regulator of lung cancer cell proliferation and radiosensitivity. LNC EBLN3P was found to interact with Nrf2, inhibiting the Keap1 and Nrf2 system’s association. Inhibition of LNC EBLN3P resulted in increased ROS production and mitochondrial injury in NSCLC cells. These exciting findings suggest that targeting LNC EBLN3P could offer a promising avenue for developing novel radiosensitizers to combat radiotherapy resistance in NSCLC.

However, some improvements below are suggested.

  1. RNA-seq and GO pathway enrichment analyses were done to explore the genes regulated by EBLN3P. If RNA-seq was done as mentioned on the manuscript, then please provide the methods of its preparation and data analysis. Also, resolution of Fig 3A could be improved.

Thank you so much for your suggestion. The methods of preparation and data analysis for the RNA-seq have been supplemented in the Materials and methods section, which are marked in red. Besides, the resolution of Fig 3A has been improved.

  1. LNC EBLN3P and lncRNA EBLN3P are both are being used interchangeably. Shouldn’t we follow one format for the consistency?

Thank you so much for your kind reminder. The “lncRNA EBLN3P” has been revised as “LNC EBLN3P” all through the manuscript.

Round 2

Reviewer 1 Report

I thank the authors for addressing all my comments. The manuscript has been improved.

Author Response

Thank you for your help!

Reviewer 2 Report

The authors have addressed majority of the questions raised by the reviewer and the manuscript is very well written and articulated. However, following queries still remain.

1. The authors pointed out by NAC experiment that oxidative stress is important. But did not justify or address the question about mitophagy. The cell has a very prominent mechanism to get rid of damaged mitochondria by mitophagy. How are these damaged mitochondria cleared by the cell is the clearance also abnormal hence the damaged mitochondria is getting accumulated leading to ROS production thereby inducing oxidative stress in these cell upon radiation exposure? The authors should provide some evidence by performing mitophagy or lysosomal function experiment to show the same. It will provide better in-depth understanding of the pathway and will be critical for better readership.

2. It is of paramount importance that the nuclear/cytosolic lysates (Fig.3) be run side by side on same gel to appreciate the difference in nuclear/cytosolic localization of Nrf2, where it should clearly show that nuclear fraction should not have actin whereas the cytosolic fraction should not have Lamin. This is a major flaw in establishing a control and appropriate study design in such an experiment.

Round 3

Reviewer 2 Report

The authors have suitably addressed all the queries.